# The End or a New Era of Development of SARS-CoV-2 Virus: Genetic Variants Responsible for Severe COVID-19 and Clinical Efficacy of the Most Commonly Used Vaccines in Clinical Practice

**DOI:** 10.3390/vaccines11071181

**Published:** 2023-06-30

**Authors:** Dimitrina Miteva, Meglena Kitanova, Hristiana Batselova, Snezhina Lazova, Lyubomir Chervenkov, Monika Peshevska-Sekulovska, Metodija Sekulovski, Milena Gulinac, Georgi V. Vasilev, Luchesar Tomov, Tsvetelina Velikova

**Affiliations:** 1Department of Genetics, Faculty of Biology, Sofia University “St. Kliment Ohridski”, 8 Dragan Tzankov str., 1164 Sofia, Bulgaria; d.georgieva@biofac.uni-sofia.bg (D.M.); m.kitanova@uni-sofia.bg (M.K.); 2Department of Epidemiology and Disaster Medicine, University Hospital “Saint George”, Medical University, 6000 Plovdiv, Bulgaria; dr_batselova@abv.bg; 3Pediatric Department, University Hospital “N. I. Pirogov,” 21 “General Eduard I. Totleben” Blvd, 1606 Sofia, Bulgaria; snejina@lazova.com; 4Department of Healthcare, Faculty of Public Health “Prof. Tsekomir Vodenicharov, MD, DSc”, Medical University of Sofia, Bialo More 8 str., 1527 Sofia, Bulgaria; 5Department of Diagnostic Imaging, Medical University Plovdiv, Bul. Vasil Aprilov 15A, 4000 Plovdiv, Bulgaria; lyubo.ch@gmail.com; 6Department of Gastroenterology, University Hospital Lozenetz, 1407 Sofia, Bulgaria; mpesevska93@gmail.com; 7Medical Faculty, Sofia University St. Kliment Ohridski, 1407 Sofia, Bulgaria; metodija.sekulovski@gmail.com; 8Department of Anesthesiology and Intensive Care, University Hospital Lozenetz, 1 Kozyak str., 1407 Sofia, Bulgaria; 9Department of General and Clinical Pathology, Medical University of Plovdiv, Bul. Vasil Aprilov 15A, 4000 Plovdiv, Bulgaria; mgulinac@hotmail.com; 10Clinic of Endocrinology and Metabolic Disorders, UMHAT “Sv. Georgi”, 4000 Plovdiv, Bulgaria; vvasilev.georgi@gmail.com; 11Department of Informatics, New Bulgarian University, Montevideo 21 str., 1618 Sofia, Bulgaria; luchesart@gmail.com

**Keywords:** COVID-19, SARS-CoV-2 variants, vaccines, effectiveness, efficacy, safety

## Abstract

Although the chief of the World Health Organization (WHO) has declared the end of the coronavirus disease 2019 (COVID-19) as a global health emergency, the disease is still a global threat. To be able to manage such pandemics in the future, it is necessary to develop proper strategies and opportunities to protect human life. The data on the SARS-CoV-2 virus must be continuously analyzed, and the possibilities of mutation and the emergence of new, more infectious variants must be anticipated, as well as the options of using different preventive and therapeutic techniques. This is because the fast development of severe acute coronavirus 2 syndrome (SARS-CoV-2) variants of concern have posed a significant problem for COVID-19 pandemic control using the presently available vaccinations. This review summarizes data on the SARS-CoV-2 variants that are responsible for severe COVID-19 and the clinical efficacy of the most commonly used vaccines in clinical practice. The consequences after the disease (long COVID or post-COVID conditions) continue to be the subject of studies and research, and affect social and economic life worldwide.

## 1. Introduction

The coronavirus disease 2019 (COVID-19) pandemic is a global pandemic caused by severe acute coronavirus 2 syndrome (SARS-CoV-2). Although “COVID-19 is now an established and ongoing health issue which no longer constitutes a public health emergency of international concern”, according to the WHO International Health Regulations Emergency Committee, it continues to have major health, economic, and social consequences worldwide [1,2]. Despite stringent control and measures having been implemented globally, and the application of innovative vaccines, the pandemic persisted for a long time (almost three years) because of the development of numerous SARS-CoV-2 variants with enhanced transmission and immune evasion [3].

At the beginning of the pandemic, there were three types of SARS-CoV-2 variant classification and definition by the Center for Disease Control and Prevention (CDC)—Variants of Interest, Variants of Concern, and Variants of High Consequence [4]. To avoid confusion and geographical stigmas, the World Health Organization (WHO) proposed that the coronavirus variants receive Greek names—Alpha, Beta, Gamma, Delta etc., from the Greek alphabet. These names will not replace the scientific labels but help the public and non-experts to differentiate between them.

Pandemic prevention measures such as the use of masks, physical and/or social distancing, the testing of symptomatic individuals, and contact tracing have proven only somewhat successful in preventing the transmission of the virus. Therefore, rapid vaccine development seemed to be the best possible option to minimize the morbidity and mortality associated with COVID-19. It turned out that the early approval of the vaccines was able to play a critical role in controlling the COVID-19 pandemic.

Although vaccines have saved millions of lives, early approval requires rigorous evaluation of their safety and efficacy [5]. The lack of data on mRNA vaccines for use on humans and their approval without long-term clinical follow-up is justified because the first clinical trials demonstrated their efficacy and satisfactory safety, according to the established and accepted criteria [6].

It is known that developing and administering a vaccine usually takes many years. The approved RNA vaccines against COVID-19 are inexpensive and easily adaptable, and their production process offers flexibility. One of the best benefits of RNA platform technology is the ability to change nucleotide sequences under the same conditions in the same place of manufacture to combat emerging mutants that evade immune system responses. Thus, mRNA vaccines may provide broad protection against different SARS-CoV-2 VOCs (variants of concern) [7].

Aside for vaccination, different therapeutic techniques such as immunotherapy and antiviral medications were and still are among the strategies to prevent infection and successfully limit the virus’s spread. Current forms of vaccinations that are used to prevent and cure COVID-19 include inactivated, viral vector, DNA, and mRNA vaccines [8]. Even last fall, two nasal COVID-19 vaccination formulations were approved for use in India and China, although their approval has not yet been applied for in Europe. These contain modified adenoviruses that are self-attenuating and, according to the results of the Phase 1 clinical trial, they provide strong protection against SARS-CoV-2 infection [9,10]. In addition, mucosal vaccines against COVID-19 continue to be discussed and developed as potential players in controlling the pandemic [11,12,13].

The number of vaccines in pre-clinical development is 199, and in clinical development—183 [14]. Currently, more than 10 vaccines against COVID-19 have been authorized under an emergency use authorization (EUA), such as Pfizer-BioNTech’s BNT162b2, Moderna’s mRNA-1273, AstraZeneca’s ChAdOx1 nCoV-19, Janssen’s Ad26.COV2.S, Sinovac’s vaccine, Sinopharm’s vaccine, CanSino’s Ad5-nCoV vaccine, ZF2001, Sputnik V’s vaccine, the EpiVacCorona vaccine, CureVac, and BBV152 [3,15,16]. Current data on available vaccines against COVID-19 and their efficacy in clinical trials are discussed below.

As of 22 May 2023, over 13 billion vaccine doses have been administered [17]. However, 70% of the world’s population has received at least one dose, in contrast to only 29.9% of low-income countries’ populations [18].

The high worldwide incidence increases the viral burden of SARS-CoV-2 in populations, raising the likelihood of new mutations. Mutations in viruses are obtained at different levels and depend on various factors: the cellular environment, the replication mechanism, polymerase enzyme action, the ability of a virus to correct mismatches by proofreading or post-replicative repair, regulation, etc. [19,20,21,22]. SARS-CoV-2 also mutates for different reasons that are related to its natural environment and external selective pressure. Natural selection plays a vital role in the occurrence of new mutations. Mutations that amplify the reproduction, transmission, and immunity escape response could continue to increase if they help the viruses survive [10]. 

More than 4000 mutations have been identified in the spike protein of the SARS-CoV-2 genome since the beginning of the COVID-19 pandemic. Studies of its mutation quantities among geographical areas were performed to analyze samples of amino acid sequences (AAS) for envelope (E), membrane (M), nucleocapsid (N), and spike (S) proteins [23,24]. The data showed that 96.40% of E AASs have no mutation, for M AASs—36.76%, for N protein—2.20%, and for C AAS—2.11%. An analysis for the presence of one mutation showed E—3.56%, M—59.64%, N—5.68%, and C—26.86%, respectively. Two mutations were found in E in 0.02%, M—2.80%, N—7.11%, and C AASs—26.15% [25].

The schematic diagram below shows the SARS-CoV-2 virus particle structure and genome organization with an arrangement of various non-structural, structural, and accessory genes: 5′-cap-UTR-replicase-S-E-M-N-3′UTR-poly (A) tail with accessory genes interspersed among the structural genes are illustrated in Figure 1. 

Most mutations that are necessary for virus transmission are in the S-gene. The spike glycoprotein (S) plays a crucial role in SARS-CoV-2 overcoming the species barrier and interspecies transmission from animals to humans; therefore, it is the primary target of selective pressure [25]. The number of mutations in the S protein of SARS-CoV-2 VOCs that occur is also illustrated in Figure 1.

Accurate information on viral mutation rates may be crucial to selecting the right strategies for controlling the pandemic and stopping virus transmission. The new variants are an expected part of the virus evolution, but monitoring each new one that surfaces is essential, especially if it is highly transmissible, vaccine-resistant, and able to cause more severe disease compared with the original virus strain.

## 2. Genetic Variants of SARS-CoV-2 Responsible for Severe COVID-19, Higher Mortality and/or Increased Transmission and Morbidity

### 2.1. Alpha Variant—B.1.1.7—United Kingdom/Kent Variant

In the fall of 2020, the United Kingdom reported a new, genetically different phylogenetic cluster of SARS-CoV-2. These new VOCs of SARS-CoV-2, including B.1.1.7 (Alpha variant according to the WHO), have a vast number of mutations [27] that affect the virus’s function, transmission, and immune system escape [28,29,30,31]. For example, the Alpha variant includes 17 mutations, 14 non-synonymous point mutations, and 3 deletions. Nine are in the Spike (S) protein, which the virus uses to penetrate cells [32]. Among these, N501Y at the RBD enhances the virus’s binding to the angiotensin-converting enzyme-2 (ACE2) receptor [28], P681H increases the transmission [33], and the deletion H69/V70 in the S protein is linked to immune escape [34]. This variant is currently known to have increased mortality, transmissibility, and potentially increased severity based on hospitalizations [28,32,35,36]. However, evidence shows that it does not affect the patient’s susceptibility to EUA monoclonal antibody treatments and minimizes neutralization by convalescent and post-vaccination sera [37,38,39,40,41,42,43]. 

Regarding infectiousness, the Alpha variant was believed to be 30–50% more contagious than the original SARS-CoV-2 strain. A study published by the CDC showed that Alpha comprised 66% of cases before the Delta variant became predominant [26].

### 2.2. Beta Variant—B.1.351—South African Variant 

The B.1.351 variant (Beta variant) was first detected in Nelson Mandela Bay, Eastern Cape Province of South Africa. It has 21 mutations and can attach more readily to human cells. Nine mutations are in the S protein; some are the same as in the B.1.1.7 variant [44,45]. Despite the similarities between these variants, the evidence suggests that they arose independently. The substitution at position 484 (E484K) in the RBD of the S protein is present in some VOCs and is reported to be “associated with escape from neutralizing antibodies” [46,47,48]. 

In early 2021, the researchers used B.1.351 in serum from people vaccinated with the Pfizer/BioNTech (BNT162b2) or Moderna (mRNA-1273) vaccines. They found that antibodies in that serum showed reduced neutralizing activity against the mutant, compared with their activity against the original virus [46,49,50]. The transmissibility is established of this variant to be elevated with potentially increased severity and hospitalizations [34,42]. 

### 2.3. Gamma Variant—P.1 or B.1.1.28—Japan/Brazilian Variant 

In January 2021, two new variants in Brazil were detected, P.1 and P.2. Although they share mutations with the other discovered variants, they seem to have arisen independently. Lineage P.1 (B.1.1.28 or 20J/501Y.V3), the Gamma variant according to the WHO, was first detected in four travelers arriving in Tokyo after visiting Brazil [51]. It has 17 unique amino acid changes, 10 of which are in the S protein [52]. In addition, the N501Y, K417N, and E484K mutations, also found in the Alpha and Beta variants, have been associated with enhanced affinity to human ACE2, and an increased transmissibility and immune escape reaction [27,53]. The P.1 variant caused the widespread infection in Manaus city, but the new lineage was absent in samples from March to November. 

The Gamma variant also has a reported potential to cause reinfections, and studies have shown a slight decrease in the efficacy of the currently available vaccines regarding this variant [34]. As mentioned, SARS-CoV-2 variants with the E484K mutation might escape neutralization antibodies from the convalescent plasma and could also increase the reinfection risk [42,53].

According to the WHO, the other variant, P.2, or the Zeta variant, has a notable mutation, E484K [54], but it is no longer detected in different countries or is at very low levels. 

### 2.4. Epsilon Variant—B.1.427/B.1.429—Californian Variant

At the end of 2020, a new variant of SARS-CoV-2 was reported in California [55,56,57]. Both lineages, B.1.427 and B.1.429, carry an identical set of three mutations (W152C; S13I; and L452R) in the ACE2-binding interface of the spike protein [56,57,58]. However, they differ in their additional synonymous and non-synonymous mutations [58]. The University of California reported that variant B.1.427/B.1.429 is four times less susceptible than the original coronavirus to neutralizing antibodies from the blood of people who recovered from COVID-19 and two times less susceptible to antibodies from the blood of people vaccinated with the Moderna (mRNA-1273) or the Pfizer/BioNTech (BNT162b2) vaccines [56,58].

### 2.5. Eta Variant—B.1.525—Nigerian Variant

Variant B.1.525 was identified in December 2020 in Nigeria and the UK. It has few mutations, which are the same as in the B.1.1.7 Lineage—E484K, deletions ΔH69/V70, and Δ144 in the NTD of the S protein. Together, these increase the transmissibility of SARS-CoV-2 [59]. 

The critical difference between the B.1.525 variant and other variants is the changes in the S protein, which may be able to attach itself to human cells more effectively in this variant. The B.1.525 has unique S protein mutations such as Q677H, Q52R, A67V, and F888L [59,60].

### 2.6. Iota Variant—B.1.526—New York Variant

In October 2021, a new variant of SARS-CoV-2, known as B.1.526, was identified in New York City [61,62,63]. The variants carry D614G and A701V mutations in the S protein and several novel point mutations [61,64]. The E484K mutation was also observed in the Iota variant and played a critical role in the loss of the activity of neutralizing antibodies and the convalescent and vaccine sera [41,64]. The other version of B.1.526 has an S477N mutation that may increase its ACE2-receptor binding affinity [62,64]. 

### 2.7. Delta Variant—B.1.617.2—India Variant

Lineage B.1.617 was first identified in Maharashtra, India, in October 2020 [65]. Within a few months, the variant was detected in different countries and was named lineage B.1.617. It contains three sublineages—B.1.617.1, B.1.617.2, and B.1.617.3. On May 2021, the sublineage B.1.617.2 (Delta variant according to the WHO) was designated as a VOC because its transmissibility was estimated to be equivalent to that of the Alpha variant [66].

B.1.617.2 has a set of S protein substitutions, and several are also present in other variants of interest/concern. Two critical mutations in the RBD domain, L452R and E484Q, affect the neutralizing antibodies’ evasion [67,68,69]. It is the first strain where these two mutations were seen together.

Preliminary evidence suggested that the B.1.617.2/Delta variant has an increased risk of hospitalization compared to the B.1.1.7/Alpha variant [70]. Additionally, recent studies have shown that the P681R mutation is a specific mutation of this lineage and is responsible for the higher pathogenicity of the B.1.617.2/Delta variant [71].

According to the CDC and some research trials, this variant has a potential reduction in neutralization by some EUA monoclonal antibody treatments and slightly reduced neutralization by post-vaccination [4,38,39,47,69].

### 2.8. Mu Variant—B.1.621—Nigerian Variant

The WHO identified the Mu variant (B.1.621) as a new SARS-CoV-2 variant of interest on August 30, 2021. For the first time, it was isolated in Columbia in January 2021. The Mu variant harbors eight mutations in the S protein: T95I, YY144-145TSN, R346K, E484K, N501Y, D614G, P681H, and D950N. Several of these mutations are present in other SARS-CoV-2 variants: E484K in Beta and Gamma, P681H and N501Y are shared with Alpha, and D950N is shared with Delta. They can reduce the sensitivity against antibodies induced by natural SARS-CoV-2 infection or vaccination [37,41,42].

The recent research on the sensitivity of the Mu variant to antibodies induced by SARS-CoV-2 infection or vaccination shown that the Mu variant is 12.4-fold more resistant to sera of eight COVID-19 convalescents who were infected during the beginning of the pandemic than the original virus. The Mu variant shows a pronounced resistance to the antibodies elicited by natural SARS-CoV-2 infection and the BNT162b2 mRNA vaccine [72,73].

Further monitoring of the Mu variant was strongly suggested because of its “constellation of mutations that indicate potential properties of immune escape.”

### 2.9. Omicron Variant (B.1.1.529 Lineage)

In November 2021, a new VOC, B.1.1.529, was identified in South Africa (Botswana). It was designated as the Omicron variant by the WHO [74]. Omicron has more than 30 changes to the S protein, several overlapping with those in the Alpha, Beta, Gamma, or Delta variants [75]. Some other reported mutations are in the envelope, membrane, N-terminal domain of the S protein, nucleocapsid protein, etc. [76,77,78]. Most of these mutations are known to increase transmissibility, viral binding affinity, and antibody escape [55,79,80,81,82]. 

Initially, Omicron was divided into three lineages (BA.1, BA.2, and BA.3), but later, by mid-2022, two more were identified (BA.4 and BA.5) [72,81,82,83,84,85].

BA.1 and BA.2 share 21 mutations in the S protein. The S proteins of BA.4 and BA.5 are similar to that of BA.2 except for the additional 69–70 deletion [85]. The BA.1 subvariant shares nine common mutations in the S protein with most VOCs, suggesting the possible recombination and origin of the Omicron variant. Among these shared mutations, six common ones were found in the Alpha, three in the Beta, three in the Gamma, and two in the Delta variants [85]. Recent studies compared transmission of the Omicron and the Delta variants and found an increased susceptibility to infection with Omicron compared to Delta, regardless of the vaccination status. A higher transmissibility and immune escape of Omicron were also found compared to the Delta variant [86,87,88,89]. Vaccines are ineffective after two doses, but with triple administration/booster, effectiveness reaches over 70% [90,91].

It is impossible to describe all known variants, so we tried to summarize the information about those that contributed to severe COVID-19, higher mortality, and increasing morbidity worldwide. These variants and their important mutations [92,93,94,95,96,97,98] are listed in Table 1.

## 3. Clinical Effectiveness of the Most Commonly Used COVID-19 Vaccines in Clinical Practice and Their Efficacy in Clinical Trials

### 3.1. Effectiveness of the Most Commonly Used COVID—2019 Vaccines in Clinical Practice against Severe Disease, Hospitalization and Mortality

When the COVID-19 pandemic began in 2019, it caused high mortality and significant economic losses worldwide. However, simultaneously, the pandemic offered a huge opportunity to progress novel and different types of therapeutics, including DNA and mRNA vaccines. The following year, the world witnessed many projects to develop vaccines against SARS-CoV-2. mRNA and plasmid DNA vaccines have been preferred for development over pathogen inactivation or attenuation because the risks of working with live pathogens are much more easily eliminated [99]. In addition, the advantages of DNA vaccine technology are easy construction, high stability, and easy delivery. However, plasmid DNA can integrate into the host genome and sometimes has poor efficacy and can present an autoimmunity risk [100]. mRNAs are non-infectious and non-integrating platforms but have problems such as instability, high innate immunogenicity, inefficient delivery in vivo, and complicated delivery due to certain storage conditions [101]. However, in early 2021, the first SARS-CoV-2 mRNA vaccine was approved only a year after the COVID-19 vaccine projects were launched. A few months later, over 10 different nucleic acid-based vaccines for SARS-CoV-2 were released by the FDA (US Food and Drug Administration) and the WHO [102].

As of May 2023, 50 vaccines against SARS-CoV-2 are in early use or have been approved for application in 201 countries [103]. However, their vaccine effectiveness is still challenged by new genetic variants.

With the inclusion of the Omicron variety to the list of VOCs, five significant SARS-CoV-2 variants have been demonstrated to be highly transmissible and respond less well to existing vaccinations than the original virus. Therefore, research and clinical trials for innovative and/or modified vaccines, and high-throughput vaccine manufacturing systems, will continue to urgently combat the difficulties of SARS-CoV-2 VOCs today and in the future. The currently approved COVID-19 mRNA and vector vaccines have an efficacy from about 70% to 97% [104]. Initial data showed a reduced neutralizing antibody response even against one of the last variants which appeared, the Omicron variant [105,106,107]. Receiving the third (booster) dose has been shown to improve protection, but it wanes rapidly from the second month after vaccination [108]. In a real-world setting, a reduced effectiveness of COVID-19 vaccines against infection or mild disease with the Omicron variant was also found [109,110,111]. 

The customized COVID-19 vaccination Comirnaty Original/Omicron BA.4–5 is approved throughout the European Union. In addition to the original strain of SARS-CoV-2, a bivalent vaccination targeting the Omicron subvariants BA.4 and BA.5 was developed. As other waves of infections are expected during the cold season, this vaccine will expand the arsenal of available vaccinations to protect patients against COVID-19. In addition, an adapted vaccine targeting BA.4 and BA.5 Omicron variants and original SARS-CoV-2 were recently recommended for approval [106], but their efficacy should be evaluated in real-world settings.

The data on protection against severe COVID-19 and hospitalization are discrepant. Some studies suggest a significantly reduced efficacy against hospitalization compared to the Delta variant, even with the third dose of the vaccine [112,113]. Other studies indicate over 90% vaccine effectiveness [109,114,115]. Data on the duration of protection against severe illness were analyzed by Zeng et al. [116] in their systematic review and meta-analysis, including 11 randomized controlled trials (161,388 participants), 20 cohort studies (52,782,321 participants), and 26 case-control studies (2,584,732 cases). They demonstrated that complete vaccination was effective against known variants, with vaccine effectiveness as follows: Alpha: 88.0% (95% CI, 83.0–91.5), Beta: 73.0% (95% CI, 64.3–79.5), Gamma: 63.0% (95% CI, 47.9–73.7), Delta: 77.8% (95% CI, 72.7–82.0), and Omicron: 55.9% (95% CI, 40.9–67.0). In addition, booster vaccination was proven to be more effective against the Delta vaccine, and its effectiveness is 95.5% (95% CI, 94.2–96.5), while Omicron’s effectiveness is 80.8% (95%CI, 58.6–91.1) [117]. Furthermore, they showed that mRNA-1273 and BNT162b2 seemed to have higher effectiveness against VOCs over others. 

To the results of two other recent studies are consistent with the above, considering all of the same studies . In addition, they even reported on the vaccine effectiveness in the context of dominant variants during clinical trials and their safety [108,117].

The effectiveness of the COVID-19 vaccines in symptomatic infection of the Delta and Omicron variants is significantly reduced but remains highly effective in preventing severe disease and hospitalization [113,118,119,120]. In addition, RNA technology has allowed the manufacturers Moderna and Pfizer/BioNTech to develop and test several Beta-, Delta-, and Omicron-specific vaccines and bivalent booster vaccines [121]. AstraZeneca and Johnson & Johnson also developed vaccine candidates based on Beta and Omicron variants [122,123]. Many of them are in the process of clinical trials.

A recent study assessed the effectiveness of COVID-19 vaccines against hospitalization after positive PCR tests for Delta and Omicron variants [124]. It examines the effectiveness of vaccines after the third dose while accounting for the increase in specificity and severity of hospitalization definitions (oxygen/ventilated patients and/or intensive care). The reason for the different values is that the interpretation of the effectiveness of the COVID-19 vaccine against hospitalization is more complex. Because Omicron causes milder disease in a more significant percentage of people (mostly younger), in hospitalized patients, COVID-19 will be an incidental finding, not a cause. These “accidental” findings lead to lower vaccine effectiveness estimates against hospitalization. However, the data in this field are still being analyzed, and the correct formulas to evaluate the overall effectiveness of vaccines are being sought [124].

Another study reported the critical outcomes of confirmed COVID-19, severe COVID-19, and/or serious adverse effects for the 10 WHO-approved vaccines [125]. Mortality evidence generally lack or have very low certainty for all of the approved vaccines. The study analyzed 41 randomized controlled trials (RCTs), including homologous and heterologous vaccinations, and the booster doses effects on 3–60-year-old participants. The conclusions are that the vaccines prepare the immune system and prevent people from being infected with SARS-CoV-2 or, if they are infected, reduce severe disease. Most have minimal differences in the incidence of serious adverse effects compared to a placebo. The efficacy and safety of COVID-19 vaccines continue to be tested in the RCTs, and information is regularly updated on the different platforms [126,127,128,129].

The evidence from these and other studies [130,131,132,133,134,135,136,137,138,139,140,141,142,143,144,145,146,147,148,149] is summarized in Table 2. 

In addition, COVID-19 vaccines’ effectiveness against severe disease, hospitalization, and mortality of the most commonly used COVID-19 vaccines in clinical practice is shown in Figure 2 and Figure 3 below.

Different studies show the efficacy and safety of COVID-19 vaccines during different periods and in different nationalities [150,151]. Each study reports different characteristics for the area, the time of administration, how many doses are administered, and the immune response duration in infected and/or healthy patients. Nevertheless, they all show that efficacy and safety are maintained and are the highest priority for monitoring and follow-up over time. Furthermore, all of them prove the increased effectiveness against severe COVID-19 disease, hospitalization, and mortality over time of the predominant SARS-CoV-2 variants [152].

### 3.2. Effectiveness of the Most Commonly Used COVID—2019 Vaccines in Clinical Practice and the Association between Vaccination and SARS-CoV-2 Reinfection

Given the SARS-CoV-2 virus’s ability to mutate, studies began investigating its potential to cause reinfection. It was initially thought that people who had recovered from COVID-19 generated a strong protective immunity. Subsequently, it was found that there were many cases of reinfection. A study summarized the cases of infected individuals with different genetic variants of SARS-CoV-2, vaccination, and the risk of reinfection [153]. Other studies also investigated the association between vaccination against COVID-19 and reinfection [154,155,156]. All evidence showed that waning is seen for both types of immunity, but less so for that which is naturally induced [157]. From high to moderate natural and/or vaccination protection against reinfection was established with non-Omicron variants, and lower protection against reinfection was established with the Omicron variant [158,159]. The vaccine effectiveness against reinfection during the dominance of the Alpha, Delta, and Omicron variants shows that individuals who are no longer infected with COVID-19 and who had completed a primary vaccination (with one or two doses) had significant immune protection against SARS-CoV-2 reinfection, hospitalization, and death [160]. Although further studies are needed on the long-term protection against circulating variants, a meta-analysis provides evidence for more robust protection of hybrid versus natural immunity lasting 12 months after the last vaccination [161].

### 3.3. Effectiveness of the Approved COVID-19 Vaccines in Clinical Practice in Special Patient Groups—Children Aged 5–11 Years, Immunocompromised Patients, Pregnant Women and Infants

At the beginning of the pandemic, it became clear that, despite the many worldwide cases of SARS-CoV-2 infection, the rate of severe COVID-19 cases in children is low. Subsequently, a systematic review of SARS-CoV-2 seroprevalence in children showed that, until April 2022, about 57% of children were seropositive globally [162]. However, infection with SARS-CoV-2 can cause severe conditions, such as multisystem inflammatory syndrome in children (MIS-C) [163]. A systematic review and meta-analysis estimated the safety and efficacy of the approved COVID-19 vaccines for children aged 5–11 years [164]. They showed that mRNA vaccines were moderately effective against Omicron variant infections but likely protected against hospitalizations.

Trials conducted during the pandemic and ongoing studies to evaluate the efficacy and safety of the COVID-19 vaccines show high seroconversion rates regardless of the previous infection status [165]. However, the vaccine trials exclude pregnant women and immunocompromised groups (cancer patients, organ transplant recipients, rheumatological diseases, etc.), resulting in a lack of data on the vaccine efficacy and safety for these groups.

Immunocompromised patient groups are of great interest because the suppression or overactivation of the immune system due to the underlying disease or concomitant treatment is possible. Moreover, the available data show that such patients are at an increased risk of severe COVID-19 and death [166,167]. A systematic review and meta-analysis were conducted on the efficacy of COVID-19 vaccination in these groups of patients (immunocompromised and immunocompetent) after the first, second, and third vaccine doses [168]. The findings show that the seroconversion range after vaccination is significantly lower in immunocompromised patients. After the second dose, improved seroconversion was observed in all patient groups, even organ transplant recipients. After the third dose, seroconversion was reported among patients who did not respond to the vaccine, although the response was variable. 

The effects of COVID-19 vaccination received during pregnancy on infection and hospitalization have also been evaluated. The findings have shown that vaccination against COVID-19 during pregnancy reduces the SARS-CoV-2 infection and hospitalization without significant maternal and fetal effects [169]. Infant passive immunity through the transmission of maternal antibodies following vaccination during pregnancy has also been studied, and evidence suggests that COVID-19 vaccination during pregnancy may reduce the risk of SARS-CoV-2 infection and hospitalization in infants [170]. Kumar et al. also discussed the concerns, challenges, and management of SARS-CoV-2 during pregnancy, paying attention to the vulnerability of pregnant women [171]. 

The availability of numerous effective SARS-CoV-2 vaccinations has provided hope to billions of people after prolonged lockdowns worldwide. By studying the virus, its mutating capabilities and transmission, and the available arsenal of therapeutic strategies, we can achieve the desired herd immunity. Therefore, it is necessary to implement several different strategies to deal with future VOCs. This will help reduce morbidity, hospitalization, and mortality from infections with different viruses and prevent future pandemics. 

Recombination is a powerful mechanism for emerging new variants and is one of the main sources of viral evolution. Viruses survive by transferring gene(s) and accumulating selective harmful and/or beneficial mutations. These lead to changes in their pathogenicity, virulence, transmission, adaptation, and immune escape options from potential vaccines [79,172,173]. Therefore, understanding the emergence of new variants and their epidemic growth is critical to public health. 

In line with this, Gong et al. [3] suggested seven future strategies to combat the COVID-19 pandemic in the light of VOCs: the use of non-pharmaceutical interventions to slow down the spread of the virus; The use of antiviral therapy for moderate to severe illness to reduce the clinical manifestations; the implementation of SARS-CoV-2 variants genomic surveillance; adaptation of the vaccines available in terms of VOC; adaptation of the tests used for SARS-CoV-2 detection; re-introducing the universal BCG vaccination as beneficial during the current pandemic; stronger international coordination and cooperation in supplying vaccines to all countries. Furthermore, there is a need to monitor and vaccinate animals to control the zoonotic spreading of the virus. In addition, cocktail treatment is an effective strategy for reducing SARS-CoV-2 mutational escape [174,175]. Another method for avoiding the development of future VOCs might be targeting more conserved protein regions with a reduced likelihood of mutation in novel vaccines [176]. An ideal vaccine would be effective against all variations, be given in a single dose, be noninjectable, and avoid cold-chain restrictions.

## 4. Conclusions

It is difficult to predict with great accuracy whether there will be new variants, what they will be, and when they will appear. It is also difficult to predict whether they will be more contagious and with greater transmissibility. Therefore, the development and adaptation of different types of vaccines should continue to be funded globally.

Current evidence shows that full vaccination with COVID-19 vaccines is highly effective against the Alpha variant and moderately effective against the Beta, Gamma, and Delta variants. Data from booster vaccination (with one or two doses) show that it is more effective against the Delta and Omicron variants. In addition, vaccine effectiveness against severe disease, hospitalization, and mortality remains high. All available studies indicate the need for continued investigation of the benefits of vaccination and the duration and efficacy of post-vaccination protection, including in special patient groups, such as immunocompromised patients, pregnant women, and infants, as well as children, to reduce the transmission of SARS-CoV-2, reinfection, hospitalization, and severe COVID-19.

Aside from genomic surveillance, there is a need for more thorough research and modeling of novel variations that include clinical data to forecast the potential danger of each new variant compared to previously examined variants. Globally, processes and regulations enable quick access and the quick exchange of viral genomes, and clinical and biological materials connected to virus isolates are required. This information will help to fully understand the mutation processes and their consequences in the SARS-CoV-2 genome, and help to limit the transmission process and prevent disease. 

Whether the era of the virus is ending or the scientific knowledge of the post-COVID consequences is just beginning, we have yet to find out. The data regarding long COVID are still accumulating, and the reasons for these effects are being sought. We need more knowledge to recognize the rue long-term COVID conditions, understand the disease’s pathological and immunological mechanisms, improve screening and assessment, improve treatment, and prevent future pandemic waves. For now, a longtime solution will be to develop multivalent vaccines that can protect against all VOCs of SARS-CoV-2.

## Figures and Tables

**Figure 1 vaccines-11-01181-f001:**
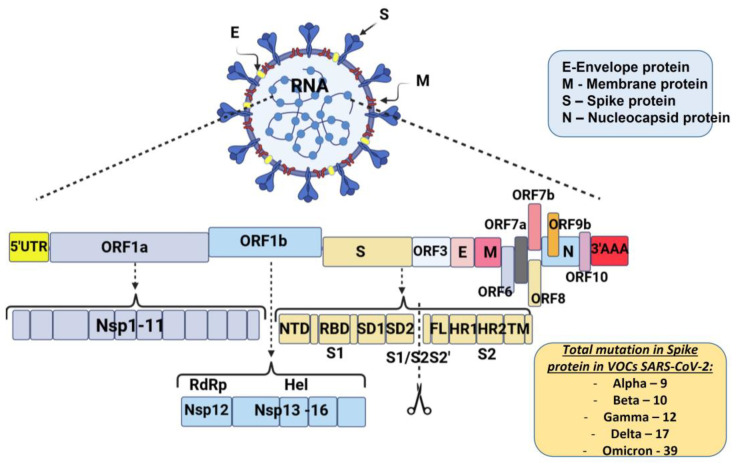
Schematic diagram of severe acute respiratory syndrome coronavirus 2 genome organization: Spike protein (S), membrane protein (M), nucleocaspid protein (N), and envelope protein (E). The genome includes open reading frames (ORFs), 16 non-structural proteins (nsp1–16) encoded by ORF1a and ORF1b, and the accessory proteins among the structural genes. S gene encodes NTD (N-terminal domain), RBD (receptor-binding domain), SD1 (subdomain 1), SD2 (subdomain 2), FL (fusion loop), HR1 (heptad repeat 1), HR2 (heptad repeat 2), and TM (transmembrane domain). Cleavage of the S1/S2 and S2′ site is shown (modified by [26]).

**Figure 2 vaccines-11-01181-f002:**
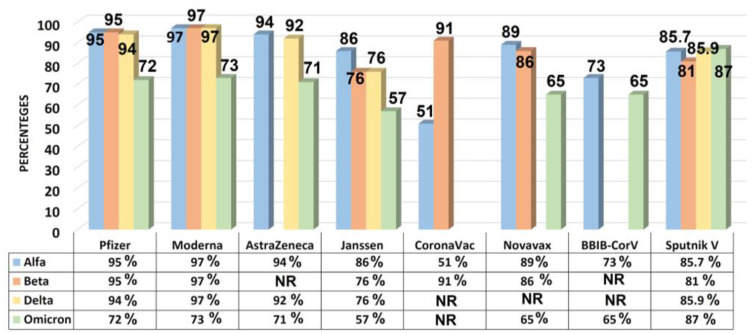
Coronavirus disease 2019 (COVID-19) vaccine effectiveness against severe disease, hospitalization, and mortality of the most commonly used COVID-19 vaccines in clinical practice. The results shown in the chart give the range of effectiveness against Alpha, Beta, Delta, and Omicron severe acute respiratory syndrome corona virus 2 variants according to data from various studies/trials conducted in 2021–2022 (NR—Not reported, VOC: variants of concern).

**Figure 3 vaccines-11-01181-f003:**
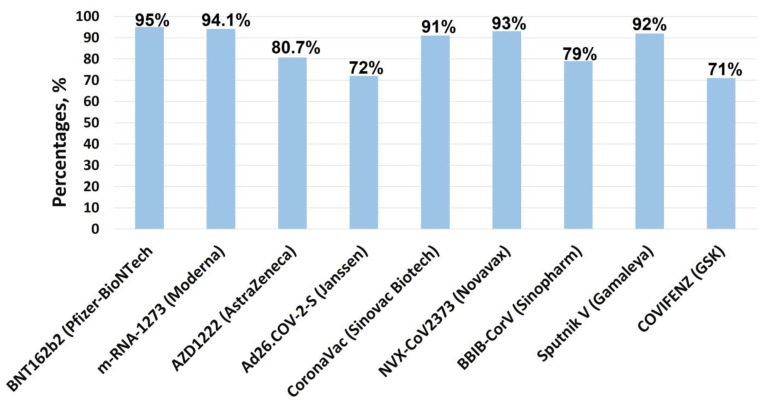
Vaccine effectiveness against infection of the most commonly used corona-virus disease 2019 vaccines against infection of SARS-CoV-2. Data in chart showed the range of effectiveness of different vaccines in different studies conducted in the period of 2021–2022.

**Table 1 vaccines-11-01181-t001:** Important mutations of severe acute respiratory syndrome coronavirus 2 (SARS-CoV-2) variants.

Variant Classifications	Name (Pango Lineage)	Spike Protein Mutations	Reference
Variants Being Monitored (VBM)	B.1.1.7 (United Kingdom variant) ALPHA	Δ69/70, Δ144Y, (E484K *) (S494P *), N501Y, A570D, D614G, P681H, T716I, S982A, etc.	Rambaut et al. [27], 2020; Liu et al. [29], 2021; Liu et al. [30], 2021; Tian et al. [31], 2021; Davies et al. [32], 2021
	B.1.351 (South Africa Variant) BETA	K417N, E484K, D80A, N501Y, D614G, D215G, L18F, 241del, 242del, 243del, A701V etc.	Karim [44], 2020; Callaway [45], 2021; Jangra et al. [48], 2021; Greaney et al. [49], 2021
	P.1 (Japan/Brazilian variant) GAMMA	K417N/T, E484K, L18F, N501Y, D614G, T20N, P26S, D138Y, R190S, H655Y, T1027I etc.	Sabino et al. [53], 2021 Voloch et al. [54], 2021; Pearson et al. [95], 2021;
	B.1.427/B.1.429 (Californian variant) EPSILON	L452R, D614G, S13I, W152C etc.	McCallum et al. [56], 2021; Tchesnokova et al. [57], 2021; Peng et al. [58], 2021
	B.1.525 (Nigerian variant) ETA	A67V, Δ69/70, Δ144, E484K, D614G, Q677H, F888L, etc.	Public Health England, 2021 (https://www.gov.uk (accessed on20 June 2023)) [66]; Ozer et al. [60], 2022
	B.1.526 IOTA	Spike: (L5F *), T95I, D253G, (S477N *), E484 *, D614G, (A701V *) etc. ORF: L3201P, T265I, Δ3675, P314L, etc.	Annavajhala et al. [61], 2021; Lasek-Nesselquist et al. [62], 2021; West et al. [63], 2021; Zhou et al. [64], 2021
	B.1.617.1 KAPPA	Spike: (T95I), G142D, E154K, L452R, E484Q, D614G, P681R, Q1071H etc.	https://www.gisaid.org/hcov19-variants (accessed on 20 June 2023) [75]; https://www.ecdc.europa.eu/en/covid-19/variants-concern (accessed on 20 June 2023) [92]
	B.1.617.3	Spike: E484Q, T19R, G142D, L452R, D950N, D614G, P681R, etc.	https://www.cdc.gov/coronavirus/2019-ncov/variants/variant-info.html (accessed on 20 June 2023) [4]
	B.1.621 MU	T95I, YY144-145TSN, R346K, E484K, N501Y, D614G, P681H, D950N	https://www.cdc.gov/coronavirus/2019-ncov/variants/variant-info.html (accessed on 20 June 2023) [4]; Collier et al. [37], 2021; Wang et al. [41], 2021; Wang et al. [42], 2021
	P.2 ZETA	Spike: E484K, D614G, V1176F; ORF: L3468V, L3930F, P314L; N: A119S, R203K, G204R, M234I etc.	Uriu et al. [72], 2021; Public Health, England, 2021, https://www.gov.uk (accessed on 20 June 2023) [66]; Pearson et al. [95], 2021
	B.1.617.2 DELTA	T19R, (G142D), 156del, 157del, R158G, L452R, T478K, D614G, P681R, D950N, T478K, W258L, 213-214del, A222V, K417N, etc.	https://www.ecdc.europa.eu/en/covid-19/variants-concern (accessed on 20 June 2023) [94] https://www.gisaid.org/hcov19-variants (accessed on 20 June 2023) [75]; Public Health England, 2021, https://www.gov.uk (accessed on 20 June 2023) [59]
Variants of Concern (VOC)	B.1.1.529 OMICRON	Δ69/70, T95I, V143del, G339D, K417N, T478K, N501Y, H655Y, N679K, L981F, Y505H, S373P, S375F, S477N, N440K, Q493R, T347K, D796Y, E484A and P681H, etc.	https://www.ecdc.europa.eu/en/covid-19/variants-concern (accessed on 20 June 2023) [4]; Aleem et al. [76], 2022; Karim et al. [78], 2021
Other variants	R.1	E484K, D614G, G769V, W152L; ORF: A2584T, P314L, G1362R, P1936H etc.	Cavanaugh et al. [96], 2021
	A.23.1	F157L, P26S, V367F, P681R, R102I, Q613H; NSP: E95K, M86I, L98F, ORF: L84S, E92K etc.	Bugembe et al. [97], 2021; Gómez et al. [98], 2021; https://www.gisaid.org (accessed on 20 June 2023) [75]
	B.1.1.318	E484K, Δ144, other mutations	Public Health England, 2021, https://www.gov.uk (accessed on 20 June 2023) [66]
	B.1.324.1	E484K, N501Y, other mutations	Public Health England, 2021, https://www.gov.uk (accessed on 20 June 2023) [66]
	P.3	E484K, N501Y, other mutations	Public Health England, 2021, https://www.gov.uk (accessed on 20 June 2023) [66]

* Detected in some sequences but not all.

**Table 2 vaccines-11-01181-t002:** Clinical efficacy of the most commonly used coronavirus disease 2019 vaccines in clinical practice against severe disease, hospitalization, and mortality.

Vaccine (Manufacturer)	Efficacy Against Infection, %	Dominant Variants During Clinical Trials	Efficacy Against VOCs and Severe Disease, %	Reference
BNT162b2 (Pfizer-BioNTech	78–95 (7 d after the second dose)	B.1, B.1.1.7 (alpha)	Alpha—95; Beta—95; Delta—94; Omicron—72 (primo vaccination); 90 (booster)	Polack et al. [130]; Thomas et al. [131]
m-RNA-1273 (Moderna)	84–94.1 (14 d after the second dose)	B.1, B.1.1.7 (alpha)	Alpha—97; Beta—97; Delta—97; Omicron—73 (primo vaccination); 90 (booster)	Baden et al. [123]; El Sahly et al. [133]
AZD1222 (AstraZeneca)	75–80.7 (14 d after the second dose)	B.1, B.1.1.7 (alpha), B.1.351 (beta)	Alpha—94; Beta—N.R; Delta—92; Omicron—71 (primo vaccination)	Voysey et al. [134]; Falsey et al. [135]
Ad26.COV-2-S (Janssen)	66–72 (28 d after the first dose)	B.1, B.1.1.7 (alpha), B.1.351 (beta)	Alpha—86; Beta—76; Delta—76; Omicron—57	Sadoff et al. [136]; Polinski et al. [137]
CoronaVac (Sinovac Biotech)	51–91	P.1 (gamma), P.2	Alpha—50; Beta—N.R; Delta—N.R; Omicron—N.R	Tanriover et al. [138]
NVX-CoV2373 (Novavax)	86–93 (7 d after the second dose)	B.1, B.1.1.7 (alpha), B.1.351 (beta)	Alpha—89; Beta—86; Delta—N.R; Omicron—65	Heath et al. [139]; Montastruc et al. [140]
BBIB-CorV (Sinopharm)	68–79	N.R	Alpha—73; Beta—N.R; Delta—N.R; Omicron—65	Al Kaabi et al. [141]; Zhang et al. [142]
WIBP-CorV (Sinopharm)	73	B.1.1.7 (alpha)	94–98.6 (hospitalization and mortality)	Al Kaabi et al. [141]; Nadeem et al. [143]
Covaxin (Bharat Biotech)	78 (14 d after the second dose)	B.1.617.2 (delta), B.1.617.1	93	Ella et al. [144]; Behera et al. [145]
Sputnik V (Gamaleya)	92	B.1.1.7 (alpha)	Alpha—85.7; Beta—81; Delta and Omicron: 85.9 (with at least one dose); 87.6 and 97.0 (for those who received more than two or three doses) two or three doses)	Shkoda et al. [146]; Matveeva et al. [147]
Convidecia (CanSino Biologics)	58–64		92–96 against severe COVID-19	Halperin et al. [148]
COVIFENZ (GSK)	71	B.1.1.7 (alpha), P.1 (gamma), B.1.617.2 (delta)	N.R	Medicago Inc. [149]

## Data Availability

Not applicable.

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
