# Peer review of "The End or a New Era of Development of SARS-CoV-2 Virus: Genetic Variants Responsible for Severe COVID-19 and Clinical Efficacy of the Most Commonly Used Vaccines in Clinical Practice"

_vaccines, 2023, doi:10.3390/vaccines11071181_

Round 1

Reviewer 1 Report

In this manuscript, Miteva and colleagues summarized the clinical consequences caused by SARS-CoV-2 variants and the clinical efficacy of the most commonly used vaccines in clinical practice. The information provided in the paper is appropriate and sufficient. Some comments below for author’s consideration:

1)    Figure 1, authors may provide more information about the mutations, such as the mutation frequency or the mutation numbers of different VOCs.

2)    Membrane Protein (M) is more commonly used than Matrix Protein.

3)    As we see the different vaccine effectiveness against SARS-CoV-2 among these vaccines, what’s the main reason for the variant performance in clinical practice. Authors may add more discussion about the advantages or special features of these vaccines.

4)    Figure 2 and 3 are not described well enough, readers may be confused by the similar title and figure legends, authors should set apart from efficacy against infection and efficacy against severe diseases.

5)    Some format mistakes. In part III, add indent in all paragraphs.  

6)    Align the X axis labeling in Figure 3. And the bar box.

7)    Other format error, in line 95, 359, 375, 377, 400   

no

Author Response

Comments and Suggestions for Authors

In this manuscript, Miteva and colleagues summarized the clinical consequences caused by SARS-CoV-2 variants and the clinical efficacy of the most commonly used vaccines in clinical practice. The information provided in the paper is appropriate and sufficient. Some comments below for author’s consideration:

  • Thank you for your time to review our paper. We acknowledge that our paper might have some issues in conformity with the referees` comments. We have addressed them and revised the manuscript accordingly. Changes are visible as highlighted and/or track changes. We sincerely thank the three reviewers for their thorough and helpful comments and suggestions. We have addressed all of the raised queries and responded to all reviewers' comments. We believe that you find these changes satisfactory, and the revisions have substantially improved the quality of the manuscript.
  • Thank you for the overall evaluation of our paper as good.

1)    Figure 1, authors may provide more information about the mutations, such as the mutation frequency or the mutation numbers of different VOCs.

  • Thank you for the valuable suggestion, we modified the figure to meet the referee`s expectations.

2)    Membrane Protein (M) is more commonly used than Matrix Protein.

  • Thank you for the valuable suggestion. Corrected.

3)    As we see the different vaccine effectiveness against SARS-CoV-2 among these vaccines, what’s the main reason for the variant performance in clinical practice. Authors may add more discussion about the advantages or special features of these vaccines.

  • Thank you for the critical point. We extended the discussion on this topic.

4)    Figure 2 and 3 are not described well enough, readers may be confused by the similar title and figure legends, authors should set apart from efficacy against infection and efficacy against severe diseases.

  • Thank you for the recommendations. We corrected the description of the figures.

5)    Some format mistakes. In part III, add indent in all paragraphs. 

  • Thank you for the suggestion. Done.

6)    Align the X axis labeling in Figure 3. And the bar box.

  • Thank you for the suggestion. Done.

7)    Other format error, in line 95, 359, 375, 377, 400  

  • Thank you for the suggestion. Done.

Reviewer 2 Report

The authors nicely summarized 1) SARS-CoV-2 variants for their associated critical mutations and clinical impacts in terms of disease severity; 2) Efficacy of the most commonly used COVID vaccines. The thorough review of published data supported two major conclusions – first conclusion: recommended COVID-19 vaccinations because: i) Full vaccination is highly effective; ii) Booster vaccination (with one or/and two doses) are more effective against Delta and Omicron; iii) Vaccine effectiveness against severe disease and mortality remains high. Second conclusion: continue genomic surveillance to monitor emergence of new variants.  The discussions presented in this review are useful and beneficial to the field.  There are some minor clarifications that need to be addressed:

1.       Page 3, line 110, “inter-specific transmission” should be “inter-species transmission”.

2.       Table 2, needs to indicate what “N.R” stands for.  "Not reported"?

3.       Proof-reading is necessary to complete and Grammarly correct some sentences.

Some sentences are incomplete and confusing.

Author Response

The authors nicely summarized 1) SARS-CoV-2 variants for their associated critical mutations and clinical impacts in terms of disease severity; 2) Efficacy of the most commonly used COVID vaccines. The thorough review of published data supported two major conclusions – first conclusion: recommended COVID-19 vaccinations because: i) Full vaccination is highly effective; ii) Booster vaccination (with one or/and two doses) are more effective against Delta and Omicron; iii) Vaccine effectiveness against severe disease and mortality remains high. Second conclusion: continue genomic surveillance to monitor emergence of new variants.  The discussions presented in this review are useful and beneficial to the field.  There are some minor clarifications that need to be addressed:

  • Thank you for your time to review our paper. We acknowledge that our paper might have some issues in conformity with the referees` comments. We have addressed them and revised the manuscript accordingly. Changes are visible as highlighted and/or track changes. We sincerely thank the three reviewers for their thorough and helpful comments and suggestions. We have addressed all of the raised queries and responded to all reviewers' comments. We believe that you find these changes satisfactory, and the revisions have substantially improved the quality of the manuscript.
  • Thank you for the overall evaluation of our paper as good.

  1. Page 3, line 110, “inter-specific transmission” should be “inter-species transmission”.
  • Thank you for the critical note. Changed.

  1. Table 2, needs to indicate what “N.R” stands for. "Not reported"?
  • Thank you for the critical note. Changed.

  1. Proof-reading is necessary to complete and Grammarly correct some sentences.
  • Thank you for the recommendations. We have proofread carefully the English grammar and style.

Reviewer 3 Report

The authors organized a manuscript entitled “SARS-CoV-2 virus variants responsible for severe COVID-19 and clinical efficacy of the most commonly used vaccines in clinical practice”. Overall, this manuscript signifies an effort to compile reports and findings regarding the SARS-CoV-2 variants and discuss the clinical efficacy of vaccines against SARS-CoV-2, based on the reality in the field. I found that the topic of current MS is adequate and acceptable for the journal’s scope. Even more, the manuscript addresses an important topic within the COVID-19 pandemic situation. The manuscript is well-prepared and it is worthy of publication. However, to this end, I have some concerns/comments related to the contents written in the manuscript.

1.   While the authors have argued for the importance of this topic, the novelty of this manuscript remains overlooked. The introduction section written in the manuscript is quite short (and lacking of persuasion) to provide insights into why this manuscript is important.

2.   The authors need to strengthen their discussion about the potency of vaccines in the mitigation of SARS-CoV-2 reinfection. Please kindly include the reference Wang et al. 2021 (http://dx.doi.org/10.1136/jim-2021-001853) and Nainu et al. 2020 (https://doi.org/10.1080/21645515.2020.1830683) in the discussion of such matter.

3.   With many variants of SARS-CoV-2 and vaccine platforms that have been described, the authors need to discuss whether special groups of patients such as the immunodeficiency patients, pregnant women, and/or kids can be protected from variants of SARS-CoV-2 by the currently available vaccine? For example, the following paper Kumar et al. 2021 (http://dx.doi.org/10.1016/j.jiph.2021.04.005) has argued the pregnant women are considered as a highly vulnerable group. These patients may have limited or impaired ability to mount adaptive immune responses, especially the the immunodeficient ones, due to impair activation of adaptive immune responses. Would this population able to mount proper immune responses after vaccinated?

4.   With limited time to produce and to test COVID-19 vaccine, the authors need to elaborate what are the criteria that need to be considered in the safety assessment of vaccine candidates? How are these criteria assessed and how long shall they be assessed prior to administration to the targeted population?

5.   All figures shall be re-prepared. The quality is not good and, in some figures such as Fig 2 and Fig 3, the figures were not labelled properly and the color of choice is not appropriate.

6.   Minor: please check for extra spaces between words. For examples, extra space the word “only” and “29.9”. Please check thoroughly in the manuscript for similar mistakes.

7.   Minor: please refrain yourself from preparing paragraphs with only one or two sentences, as shown in Lines 92-93, 130-131, 132-134, 255-256, etc. Please check thoroughly!

8.   The current form of the manuscript is not suitable for publication. I highly believe that the manuscript needs to be proof-read carefully to check for major/minor English errors.

9.   Lastly, the similarity report demonstrates around 28% of similarity in the manuscript. Please paraphrase more to reduce the similarity.

Finally, I recommend that the authors shall address all comments/concerns above prior to consideration of publication in this journal.

The current form of the manuscript is not suitable for publication. I highly believe that the manuscript needs to be proof-read carefully to check for major/minor English errors.

Author Response

The authors organized a manuscript entitled “SARS-CoV-2 virus variants responsible for severe COVID-19 and clinical efficacy of the most commonly used vaccines in clinical practice”. Overall, this manuscript signifies an effort to compile reports and findings regarding the SARS-CoV-2 variants and discuss the clinical efficacy of vaccines against SARS-CoV-2, based on the reality in the field. I found that the topic of current MS is adequate and acceptable for the journal’s scope. Even more, the manuscript addresses an important topic within the COVID-19 pandemic situation. The manuscript is well-prepared and it is worthy of publication. However, to this end, I have some concerns/comments related to the contents written in the manuscript.

  • Thank you for your time to review our paper. We acknowledge that our paper might have some issues in conformity with the referees` comments. We have addressed them and revised the manuscript accordingly. Changes are visible as highlighted and/or track changes. We sincerely thank the three reviewers for their thorough and helpful comments and suggestions. We have addressed all of the raised queries and responded to all reviewers' comments. We believe that you find these changes satisfactory, and the revisions have substantially improved the quality of the manuscript.
  • Thank you for the overall evaluation of our paper as good.

  1. While the authors have argued for the importance of this topic, the novelty of this manuscript remains overlooked. The introduction section written in the manuscript is quite short (and lacking of persuasion) to provide insights into why this manuscript is important.
  • Thank you for the critical notes and recommendations. We modified the introduction section based on your suggestions to provide more concise goal and hypothesis.

  1. The authors need to strengthen their discussion about the potency of vaccines in the mitigation of SARS-CoV-2 reinfection. Please kindly include the reference Wang et al. 2021 (http://dx.doi.org/10.1136/jim-2021-001853) and Nainu et al. 2020 (https://doi.org/10.1080/21645515.2020.1830683) in the discussion of such matter.
  • Thank you for the great recommendations. We added the proposed papers.

  1. With many variants of SARS-CoV-2 and vaccine platforms that have been described, the authors need to discuss whether special groups of patients such as the immunodeficiency patients, pregnant women, and/or kids can be protected from variants of SARS-CoV-2 by the currently available vaccine? For example, the following paper Kumar et al. 2021 (http://dx.doi.org/10.1016/j.jiph.2021.04.005) has argued the pregnant women are considered as a highly vulnerable group. These patients may have limited or impaired ability to mount adaptive immune responses, especially the the immunodeficient ones, due to impair activation of adaptive immune responses. Would this population able to mount proper immune responses after vaccinated?
  • The referee is right to pointed out that
  • We tried to cover this important topic and to discuss the vulnerability of some patients groups in the light of impaired immune responses and vaccination effectiveness.

  1. With limited time to produce and to test COVID-19 vaccine, the authors need to elaborate what are the criteria that need to be considered in the safety assessment of vaccine candidates? How are these criteria assessed and how long shall they be assessed prior to administration to the targeted population?
  • Thank you for the valuable suggestions. We cover these questions in the paper.

  1. All figures shall be re-prepared. The quality is not good and, in some figures such as Fig 2 and Fig 3, the figures were not labelled properly and the color of choice is not appropriate.
  • Thank you for the critical notes. We have corrected the figures.

  1. Minor: please check for extra spaces between words. For examples, extra space the word “only” and “29.9”. Please check thoroughly in the manuscript for similar mistakes.
  • Thank you for noticing this issue. Corrected.

  1. Minor: please refrain yourself from preparing paragraphs with only one or two sentences, as shown in Lines 92-93, 130-131, 132-134, 255-256, etc. Please check thoroughly!
  • Thank you for noticing this issue. Corrected.

  1. The current form of the manuscript is not suitable for publication. I highly believe that the manuscript needs to be proof-read carefully to check for major/minor English errors.
  • We proofread the paper again paying special attention to the English grammar and style.

  1. Lastly, the similarity report demonstrates around 28% of similarity in the manuscript. Please paraphrase more to reduce the similarity.
  • We check the similarity, and corrected that one that is outside the references.

Finally, I recommend that the authors shall address all comments/concerns above prior to consideration of publication in this journal.